# Characterization of Nigerian breast cancer reveals prevalent homologous recombination deficiency and aggressive molecular features

Jason J. Pitt et al.#

Racial/ethnic disparities in breast cancer mortality continue to widen but genomic studies rarely interrogate breast cancer in diverse populations. Through genome, exome, and RNA sequencing, we examined the molecular features of breast cancers using 194 patients from Nigeria and 1037 patients from The Cancer Genome Atlas (TCGA). Relative to Black and White cohorts in TCGA, Nigerian HR + /HER2 − tumors are characterized by increased homologous recombination deficiency signature, pervasive TP53 mutations, and greater structural variation—indicating aggressive biology. GATA3 mutations are also more frequent in Nigerians regardless of subtype. Higher proportions of APOBEC-mediated substitutions strongly associate with PIK3CA and CDH1 mutations, which are underrepresented in Nigerians and Blacks. PLK2, KDM6A, and B2M are also identified as previously unreported significantly mutated genes in breast cancer. This dataset provides novel insights into potential molecular mechanisms underlying outcome disparities and lay a foundation for deployment of precision therapeutics in underserved populations.

#A full list of authors and their affliations appears at the end of the paper.. These authors contributed equally: Jason J. Pitt and Markus Riester. These authors jointly supervised this work: Kevin P. White, Dezheng Huo, Olufunmilayo I. Olopade and Jordi Barretina.

Breast cancer is a heterogeneous disease comprising distinct subtypes. Both global burden and severity of the disease vary widely across populations, with women of African ancestry being diagnosed at a younger age, having more clinically aggressive disease and advanced stage at diagnosis, as well as having higher mortality rates than age-matched women of European or Asian ancestry[1–4]. Molecular and genetic characteristics strongly influence breast cancer prognosis and treatment, with HER2 amplification (human epidermal growth factor receptor 2 [ERBB2]) and hormone receptor (HR; estrogen receptor [ER] and progesterone receptor [PR]) expression being the best examples.

Recent large sequencing studies, for instance the International Cancer Genome Consortium (ICGC) and The Cancer Genome Atlas (TCGA), have refined our knowledge of the genomic landscape and pathogenesis of breast cancer, have provided insight into tumor evolution and mechanisms of drug resistance, and have laid a pathway to deployment of precision therapeutics[5–15]. Moreover, these large public datasets have also enhanced our understanding on the divergent mutation accretion processes; most notably in breast cancer, studies have shown high APOBEC (apolipoprotein B mRNA editing enzyme, catalytic polypeptide-like)-related mutagenesis, especially in HER2 + tumors[16], whereas BRCA1/2 mutations are strongly associated with signatures depicting DNA repair deficiency[17].

The cases used to elucidate the genetic basis of breast cancer have been overwhelmingly from women of European ancestry, which reiterates the need for data from underrepresented ethnicities[18–20]. Moreover, paucity of data from African countries potentially widens the knowledge gap that contributes to global disparities in breast cancer outcomes. To get a comprehensive understanding of the genetic architecture of breast cancer in West Africans, the founder population of a large proportion of Black women in the United States, we conducted whole-genome sequencing (WGS), whole-exome sequencing (WES), and transcriptome sequencing (RNA sequencing (RNA-seq)) on 194 tumors from Nigerian patients and performed a comparative analysis with Black women of African ancestry and White women of European ancestry from the United States in TCGA. In comparison with the TCGA cohorts, we observe that HR + /HER2 − Nigerians are enriched for molecular characteristics associated with aggressive biology. To the best of our knowledge, combined with African American patients in TCGA, this is the largest breast cancer genomics study on tumors from women of African ancestry to date.

## Results

**Study populations.** The Nigerian cohort comprised 194 breast cancer patients: 40 with WGS data, 129 with WES data, and 103 with RNA-seq data (Supplementary Fig. 1). Of the 1097 TCGA breast cancer patients with either WES ($n = 1035$) or WGS ($n = 84$), 1030 were assigned without ambiguity to 3 ancestral race groups (Black, ≥ 50% African; White, ≥ 90% European; Asian, ≥ 90% Asian ancestry) and the other 67 had mixed racial background (Supplementary Data 1a–e). DNA sequencing data from all samples was uniformly processed using the SwiftSeq workflow (manuscript in preparation). Patient clinical and pathologic characteristics are shown in Supplementary Tables 1–5. Nigerians were much younger and had more advanced stage at diagnosis than patients in the TCGA cohort, reflecting population structure and lack of screening in the country.

**Mutation landscape of Nigerians compared with Americans.** Congruous with previous studies including the Surveillance Epidemiology End Results dataset[2,21], we observed a strong enrichment of HR −/HER2 − (i.e., triple-negative for ER/PR/

HER2; 43% in Nigerian vs. 33% in Black and 13% in White) and HR −/HER2 + (25 vs. 6 and 2%) subtypes in the Nigerian cohort (Fig. 1a). PAM50 subtyping revealed a similar enrichment of Basal-like (32 vs. 35 and 15%) and HER2 enriched (29 vs. 9 and 5%) in Nigerian women (Fig. 1b).

Across all 1164 individuals—both TCGA and Nigerians—with WES data, we identified 25 genes that were significantly mutated above background (MutSigCV, $Q < 0.05$). Three of these genes (PLK2, KDM6A, and B2M; Supplementary Methods; Supplementary Fig. 2) had little or no previous evidence of harboring mutations that drive breast carcinogenesis. A fourth gene, GPS2, was also identified by Bailey et al.[22] while this manuscript was under review. Notably, mutations in PLK2 (Fisher's exact, $P = 0.05$) and KDM6A ($P = 0.06$) were enriched within HER2 + patients. Combined with previously reported significantly mutated genes in breast cancer[13,23], this resulted in 44 driver genes. These genes, along with those recurrently affected by copy number changes[6] (Supplementary Table 6), were used for gene-centric comparisons by race/ethnicity.

Consistent with the aggressive subtype composition in Nigerians, we found an enrichment of TP53 alterations (62 vs. 46 and 29%; Fisher's exact, Benjamini–Hochberg [BH] $P < 0.0001$) as well as a lower prevalence of PIK3CA mutations (17 vs. 20 and 36%; BH $P < 0.0001$) (Fig. 1c). Combined BRCA1 germline and somatic variants were also enriched in the Nigerian cohort (11.6 vs. 7.0 and 4.0%; BH $P = 0.03$). CDH1 mutation was rare in Nigerians (0.8 vs. 6.4 and 16.2%; BH $P < 0.0001$), whereas GATA3 alterations were more common in this population (17.1 vs. 10.0% and 9.5%; BH $P = 0.24$).

When comparing recurrently gained or lost regions as identified by GISTIC2 (Supplementary Fig. 3; Supplementary Methods), we found that all high confidence peaks identified in the Nigerian cohort had corresponding peaks within 10 Mb in the combined TCGA cohort. In line with immunohistochemistry (IHC) and PAM50, the ERBB2 locus (17q12) was enriched in Nigerians (amplified in 24 vs. 12 and 10%; BH $P = 0.002$), as was its wide neighboring peak at 17q23.1 (TBX2 locus, BH $P = 0.1$) (Fig. 1d).

Within IHC subtypes, significantly mutated genes and copy number peaks generally displayed similar proportions across ethnicities (Fig. 1e, f), suggesting that most mutation frequency differences reflects subtype differences across ethnicities. Within the HR + /HER2 − subtype, however, there were more TP53 and GATA3 mutations, and fewer PIK3CA and CDH1 mutations in Nigerians, compared with TCGA Blacks and Whites (all $P < 0.05$). These results are not strongly influenced by age (Supplementary Methods) and suggest that HR + /HER2 − breast cancers in Nigerian women have genomic lesions consistent with more aggressive disease.

**Mutation signatures across subtypes and driver mutations.** We next extracted breast cancer mutational signatures in the 122 WGS and 500 WES samples from Nigerian and TCGA cohorts harboring 100 or more mutations (Supplementary Methods). Of the nine independently identified signatures, signatures A (APOBEC C > T), B (APOBEC C > G), C (Aging), H (Signature 8), and I (homologous recombination deficiency [HRD]) closely matched to previously identified breast cancer signatures (Supplementary Figs. 4 and 5A). Given that these five signatures had high correlation between exomes and genomes (Supplementary Fig. 5b), we examined those in subsequent analyses. Combined, they explain the vast majority of mutations regardless of race/ethnicity (Fig. 2a) or subtype (Fig. 2b).

We observed increased contributions from APOBEC C > T (Mann–Whitney $U$ [MWU], $P = 3.5 \times 10^{-9}$) and APOBEC C > G

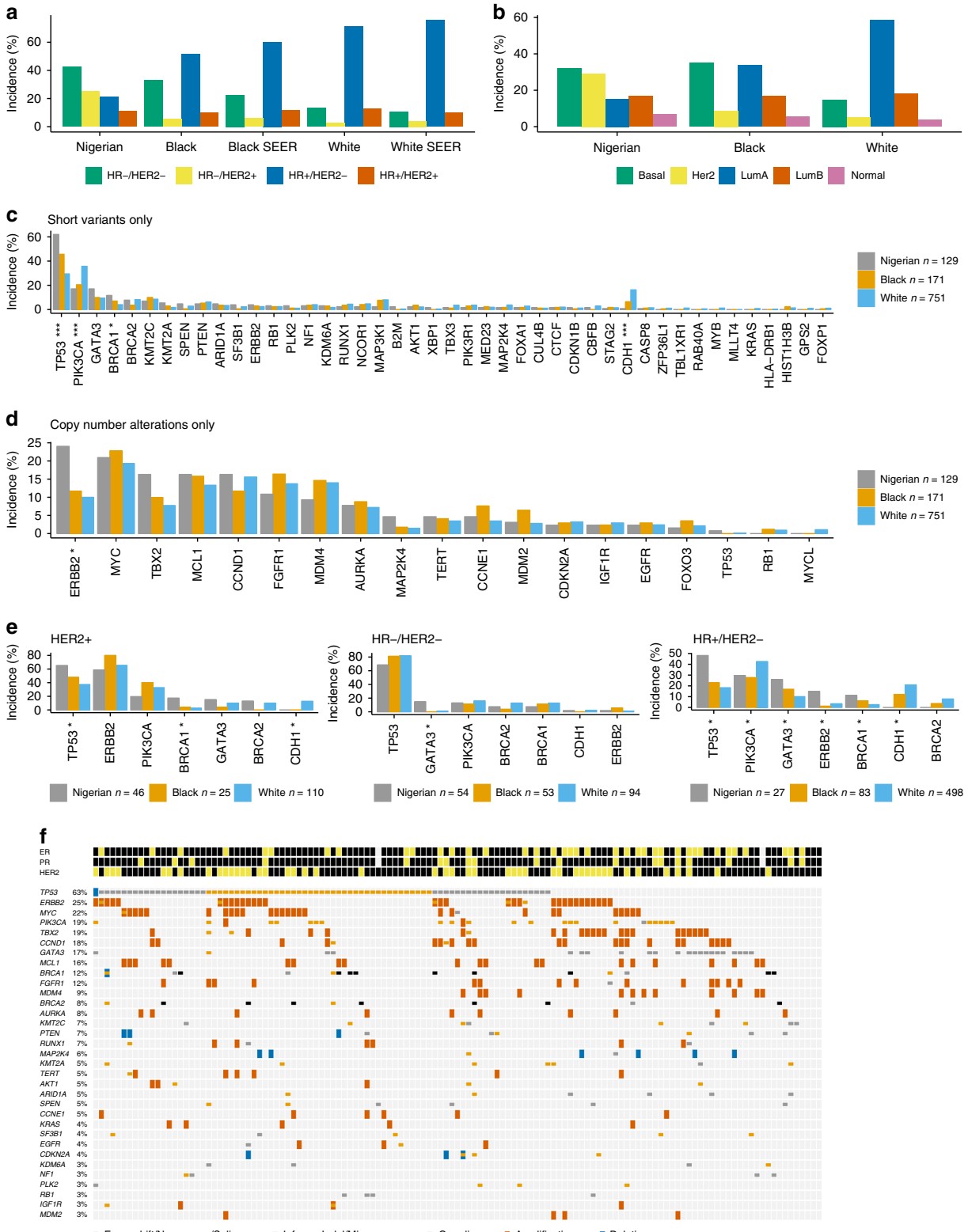

**Fig. 1** Landscape of breast cancer in Nigerians compared to Black and White Americans. **a** Proportion of IHC subtypes in the Nigerian, Black, and White cohorts from TCGA and in the SEER database. **b** Proportion of PAM50 subtypes in Nigerians, Blacks, and Whites. **c** Comparison of the frequencies of short variants (SNVs and indels) in 44 breast cancer drivers in all cohorts. **d** Alteration frequencies of 19 genes recurrently affected by CNAs (homozygous deletions and amplifications). **e** Comparison of key breast cancer drivers stratified by IHC subtype. Both short variants and copy number events are included. **f** Oncoprint of short mutations and CNAs in Nigerians. Recurrently mutated genes that were altered least 3% of Nigerians are shown. *P < 0.05; **P < 0.001; ***P < 0.0001 (Fisher's exact with P-values adjusted via the Benjamini–Hochberg method)

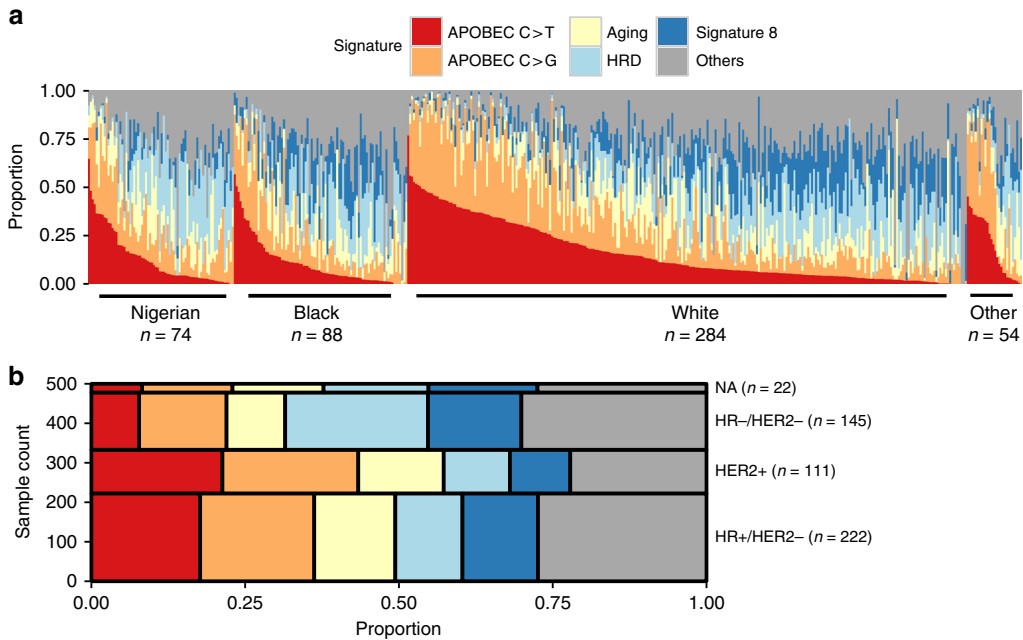

**Fig. 2** Mutation signature contributions across race/ethnicity and subtype. **a** The contribution (proportion) of mutation signatures (Signatures D, E, F, and G are combined into "Other") within each individual. Individuals are partitioned by race/ethnicity and ordered by APOBEC C > T signature contributions (high to low). The number of individuals representing each cohort is shown. **b** Mekko plot of the proportional contributions of mutation signatures across IHC subtypes

($P = 0.044$) signatures in HR + tumors compared with HR − tumors, which is consistent with previous findings[24,25]. On average, the HRD signature was more active in HR − tumors ($P = 2.2 \times 10^{-15}$) (Supplementary Fig. 6a–d). Consistent with previous work[16], HER2 + tumors had the highest contributions from APOBEC C > T and C > G signatures ($P = 1.6 \times 10^{-8}$ and $P = 9.1 \times 10^{-4}$, respectively) (Fig. 2b; Supplementary Fig. 6e, f). Similarly, we recapitulated the known aging signature associations (Supplementary Methods) and confirmed higher HRD contributions in individuals harboring deleterious germline or somatic *BRCA1/2* mutations ($P = 6.2 \times 10^{-7}$)[17].

*TP53* mutations were associated with higher HRD contributions (MWU, $P = 2.1 \times 10^{-13}$), higher missense mutation burden ($P = 6.5 \times 10^{-45}$), and increased copy number segmentation ($P = 2.0 \times 10^{-43}$) (Fig. 3a; Supplementary Methods). In contrast, *CDH1* or *PIK3CA* mutations—which frequently co-occur ($P = 3.8 \times 10^{-8}$)—were associated with lower HRD contributions (*CDH1* $P = 5.2 \times 10^{-11}$; *PIK3CA* $P = 2.1 \times 10^{-17}$) in addition to higher contributions from APOBEC C > T ($P = 3.0 \times 10^{-9}$; $P = 3.8 \times 10^{-17}$) and C > G ($P = 1.7 \times 10^{-4}$; $P = 2.1 \times 10^{-6}$) (Fig. 3a). Importantly, these significant associations persisted even when considering only HR + /HER2 − tumors (Fig. 3b). These findings suggest a consistent interplay between driver mutations and the relative activity of mutational processes.

**Mutation signatures across race and ethnicity**. Signature 8 demonstrated substantial contribution differences between cohorts. This effect was the most pronounced in HR −/HER2 − tumors, where Nigerians and Blacks ($P = 4.4 \times 10^{-6}$), Nigerians and Whites ($P = 4.6 \times 10^{-12}$), as well as Blacks and Whites ($P = 0.023$) were significantly different from one another (Fig. 4a). Notably, Whites presented with remarkably higher signature 8 in HR −/HER2 − (mean = 20.6%) compared with HR + /HER2 − (mean = 12.2%) tumors ($P = 3.4 \times 10^{-7}$), which was recapitulated using WGS data ($P = 6.9 \times 10^{-3}$) (Supplementary Fig. 8a, b). These subtype differences were not observed for either Nigerians or Blacks.

In the HR + /HER2 − subtype, the APOBEC C > T signature displayed differences by race/ethnicity with Nigerian and Black cohorts having lower APOBEC C > T contributions compared with Whites (MWU, $P < 0.05$). In the HR −/HER2 − subtype, Nigerians had increased APOBEC C > G signature relative to the Black and White cohorts ($P < 0.05$) (Supplementary Fig. 7a, b). Strikingly, HR + /HER2 − Nigerian tumors had higher HRD signature contributions compared with both Black ($P = 1.8 \times 10^{-4}$) and White ($P = 1.6 \times 10^{-4}$) cohorts (Fig. 4b). This finding was confirmed using data from WGS (Supplementary Fig. 8c). Structural variants (SVs) are more prevalent in tumor types with HRD defects such as ovarian and basal-like breast cancers[13,26]. In this same set of genomes, Nigerians had more SVs than both Black (MWU, $P = 0.03$) and White cohorts ($P = 2.8 \times 10^{-4}$). Similar with the HRD signature, SVs counts in HR + /HER2 − Nigerians (~551 SVs per genome) were reminiscent of HR −/HER2 − (~626 SVs per genome) (Fig. 4c). Differences between Nigerians and Whites in HRD signature and SVs (both $P < 2.0 \times 10^{-3}$) extended to HER2 + cases as well (Fig. 4b, c). Taken together, multiple lines of evidence suggest that HR + /HER2 − Nigerians have increased HRD and genomic complexity compared with the Black and White cohorts. Furthermore, genome data suggests a potentially more granular stratification by African ancestry.

We postulated that increased HRD in HR + /HER2 − Nigerians may be explained by increased prevalence of *TP53* mutations as well as fewer *PIK3CA* and *CDH1* mutations—although not necessarily causatively. Using multivariate modeling (Supplementary Methods), we investigated the effect of race/ethnicity on HRD adjusting for age and missense burden, as well as mutation status in *TP53*, *BRCA1/2*, *PIK3CA*, and *CDH1*. Although many of these factors have significant, independent effects, they cannot entirely account for the racial/ethnic HRD disparities seen across HR + /HER2 − tumors.

**HRD-APOBEC signature balance**. Several threads of evidence suggest a possible interplay between the HRD and APOBEC signature contributions, particularly in HR + /HER2 − breast

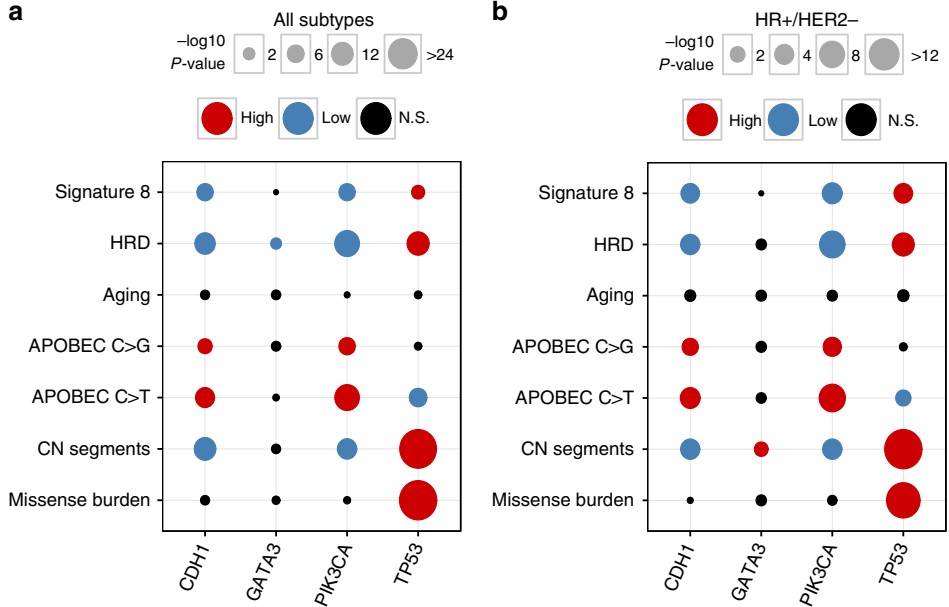

**Fig. 3** Associations between genome-wide oncogenic features and the mutation status of common driver genes. Dot plot depicting the relationships between mutation status in *TP53*, *PIK3CA*, *CDH1*, and *GATA3*, and mutation signatures (APOBEC C > T, APOBEC C > G, aging, HRD, and signature 8), missense mutation burden, and copy number (CN) segments **a** across all IHC subtypes ($n = 500$) and **b** within HR +/HER2 − ($n = 222$). Only TCGA data, including samples lacking mutation signature estimates, was used for CN associations (all subtype $n = 1,023$; HR +/HER2 − $n = 635$). No samples were excluded based on race/ethnicity. Comparisons between mutation status and genomic features were performed with Mann–Whitney U and P-values were corrected for multiple testing (Benjamini–Hochberg method). Circle size is proportional to the magnitude of the − log10 BH P-values (i.e., lower BH P-values have larger circles). If mutation status associated with a significant increase or decrease of a genomic feature, the corresponding circle is colored red or blue, respectively. Non-significant (NS) comparisons are colored black

cancers: (1) we identified racial/ethnic differences in mutation prevalence for *TP53*, *CDH1*, and *PIK3CA*; (2) we found associations between these mutations and mutation signatures (Fig. 3a); and (3) consistent with differential mutation status, HRD activity was increased in Nigerians, whereas APOBEC C > T displayed reduced activity in Nigerians and Blacks compared with Whites (Supplementary Fig. 7a). Furthermore, within this subtype, HRD had a notable negative correlation with both the APOBEC C > T ($\rho = -0.56$, permutation test $P < 0.0001$; Supplementary Methods) and APOBEC C > G ($\rho = -0.30$, $P < 0.0001$) signatures. Integrating these findings, we postulated that a balance of HRD and APOBEC signature contributions exists and can be discriminated—if not dictated—by mutations in *BRCA1/2* (germline and somatic), *TP53*, *PIK3CA*, and *CDH1*. For each tumor, we combined APOBEC C > T and C > G contributions, and plotted them against that of HRD (Fig. 5a). Tumors were partitioned based on the presence of *CDH1* or *PIK3CA* mutations (*CDH1/PIK3CA*), *TP53* or *BRCA1/2* mutations (*TP53/BRCA1/BRCA2*), mutations from both aforementioned categories (Both), or mutations in neither of the aforementioned categories (Neither). APOBEC contributions were significantly higher in *CDH1/PIK3CA* compared with the *TP53/BRCA1/BRCA2* (Dunn's test, $P = 1.8 \times 10^{-6}$) and Neither ($P = 7.2 \times 10^{-9}$) groups. Tumors harboring mutations from both groups (Both) had lower APOBEC contributions than *CDH1/PIK3CA* ($P = 0.11$), yet higher than *TP53/BRCA1/BRCA2* ($P = 5.0 \times 10^{-3}$) (Fig. 5b). In contrast, *TP53/BRCA1/BRCA2* had significantly higher HRD contributions than all other groups ($P$ *CDH1/PIK3CA* $= 9.9 \times 10^{-15}$; Both $= 1.6 \times 10^{-4}$; Neither $= 2.1 \times 10^{-4}$), whereas *CDH1/PIK3CA* had significantly lower contributions than all other groups ($P$ Both $= 3.5 \times 10^{-3}$; Neither $= 1.2 \times 10^{-3}$) (Fig. 5c). These findings were similar when considering all samples simultaneously (Supplementary Fig. 9a–c).

The signature patterns for the Neither group most closely resembled those of *TP53/BRCA1/BRCA2* (Fig. 5a–c), suggesting that there may be other mechanisms, such as inactivation of other homologous recombination genes[27] or *BRCA1/2* methylation[28], which promote increased HRD activity. When looking at the proportion of these mutational groups across HR +/HER2 − samples (including those without signature estimates), the groups with the highest HRD and lowest APOBEC—*TP53/BRCA1/BRCA2* and Neither—encompassed 70.3% Nigerians and 66.3% Blacks but only 47.7% of Whites ($\chi^2$-test, $P = 1.2 \times 10^{-3}$) (Fig. 5d). This suggests that individuals with African ancestry are more likely to fall within mutational groups associated with increased HRD and lower APOBEC contributions. Consistent with this assertion, the HR +/HER2 − Black cohort had greater copy number segmentation (MWU, $P = 0.022$), more structural variation (Dunn's test, $P = 0.028$), and increased HRD in WGS (Dunn's test, $P = 0.015$) compared with Whites (Fig. 4b; Supplementary Fig. 8c). Throughout African ancestry tumors, prevalent aggressive and limited favorable molecular features could in part explain known racial/ethnic mortality disparities within the HR +/HER2 − subtype[29]. This has significant clinical implications, because HRD tumors are more likely to be sensitive to platinum-based chemotherapy, PARP (poly (ADP-ribose) polymerase) inhibition, and immunotherapy[28].

**Infiltrating immune cell inference by RNA signatures**. Given the high HRD signature activity and the fact that DNA repair gene alterations have been linked to checkpoint inhibitor efficacy, we next investigated gene expression signatures related to immune cell infiltration, or immune signatures, with RNA-seq (Fig. 6a and Supplementary Table 7). Most immune signatures— B-cell, Cytotoxic T cell, Fibroblast, Interferon (IFN)-γ, Type I

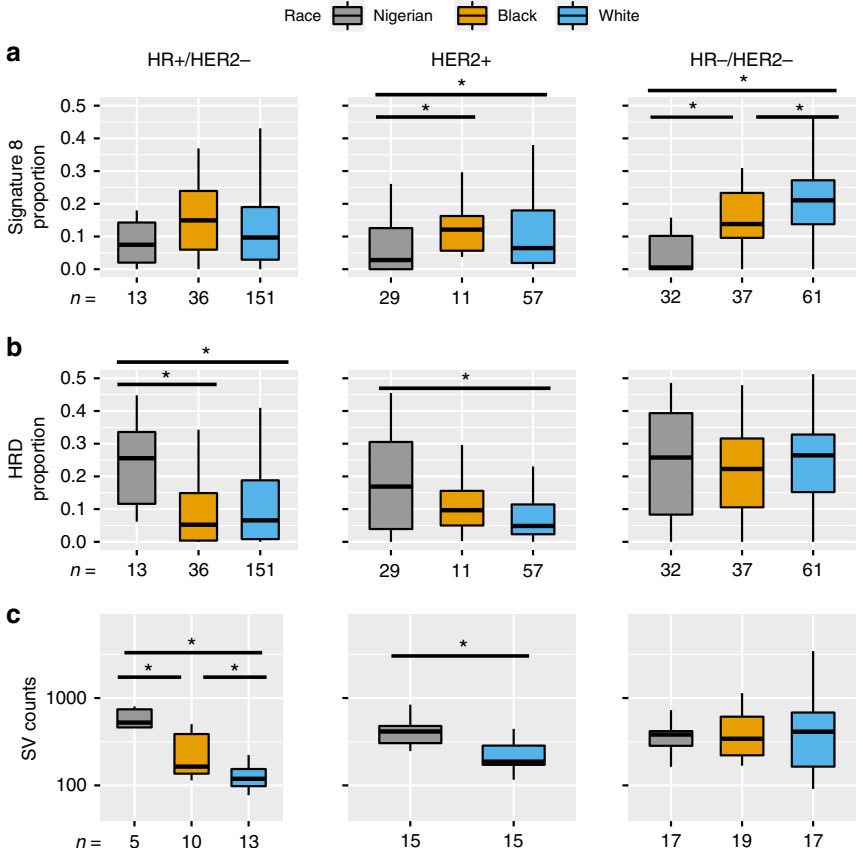

**Fig. 4** Mutation signature contributions and structural variant counts by race/ethnicity and IHC subtype. Mutation signature contributions from **a** signature 8 and **b** HRD subdivided by race/ethnicity and IHC subtype. **c** Boxplots representing the number of SVs identified across WGS samples partitioned by race/ethnicity and IHC subtype. Asterisks denote significant differences ($P < 0.05$) between groups using Kruskal–Wallis tests followed by post-hoc comparisons with Dunn's test. Each box represents the upper and lower quartiles of the data, and the median is depicted with a horizontal line. Upper and lower whiskers extend to the largest and smallest values within [1.5 × interquartile range], respectively

IFN, and Proliferation—displayed statistically significant differences across PAM50 subtypes (analysis of variance, all $P < 0.0001$; Supplementary Methods). Racial differences adjusted for PAM50 subtype, however, were modest (Fig. 6b and Supplementary Fig. 1,0). The Cytotoxic cell signature ($P = 0.004$) was lower in Nigerians in all subtypes but Basal, whereas the Fibroblast signature ($P = 0.01$) was consistently highest in Nigerians. Type I IFN signature scores ($P = 0.01$) were enriched in Luminal subtypes for both Nigerians and Blacks, which potentially indicates that tumors from these racial groups would respond better to immunotherapy[30]. Lastly, macrophage infiltration in Nigerians was highest in the Basal subtype, similar to what has been reported in other studies, including one in a small subset of Nigerian patients[31,32].

We next tested these immune signatures for association with potential predictors of response to immunotherapy. We considered the combined APOBEC C > T and C > G, and the HRD mutation signatures as the two independent mutational processes generating putative neoantigens, as well as mutation burden and chromosomal instability (CIN)[33–35]. APOBEC mutation signature contribution was positively correlated with mutation burden ($\rho = 0.35$, Spearman's rank correlation, BH $P < 0.0001$). Consistent with recent reports, we found APOBEC contribution being further associated with increased T-cell infiltration ($\rho = 0.25$, BH $P < 0.0001$) and CIN being positively correlated with mutation burden ($\rho = 0.28$, BH $P < 0.0001$), while negatively correlated with T-cell infiltration ($\rho = -0.08$, BH $P < 0.01$)[34,35]. The same trends were observed in the Nigerian and TCGA cohorts

separately with similar effect sizes (Fig. 6c, d), although, in the former, most were not significant after multiple testing correction potentially due to the smaller sample size (Fig. 6a).

## Discussion

To date, this study is the largest genomic analysis of breast cancer among women of African ancestry. Aggressive molecular subtypes were found to be more prevalent in Nigerian patients, which has been consistently documented in breast tumors across West Africa[2]. The extent to which this disparity represents disparate biology, environmental influences, or a combination thereof remains unknown. Recently, ER expression was demonstrated to be a heritable trait in breast cancer[36], suggesting that genetically influenced basal expression levels may contribute to subtype differentiation. Given that genetic background associates with phenotypes relevant to breast cancer, it is reasonable to postulate that patterns of somatic mutations may differ across genetically distinct populations. Here we have shown that regardless of subtype, aggressive molecular features are prevalent in breast tumors from Nigerian women.

Including Nigerian samples along with TCGA allowed us to identify *PLK2*, *KDM6A*, and *B2M* as novel significantly mutated genes in breast cancer, with the former two enriched in the HER2 + subtype. *PLK2* is a cell cycle regulator and presumed tumor suppressor, whereas *KDM6A* is a chromatin modifier frequently mutated in other cancer types (e.g., pancreatic, esophageal, and bladder)[37–40]. *B2M* inactivation was recently reported to be a

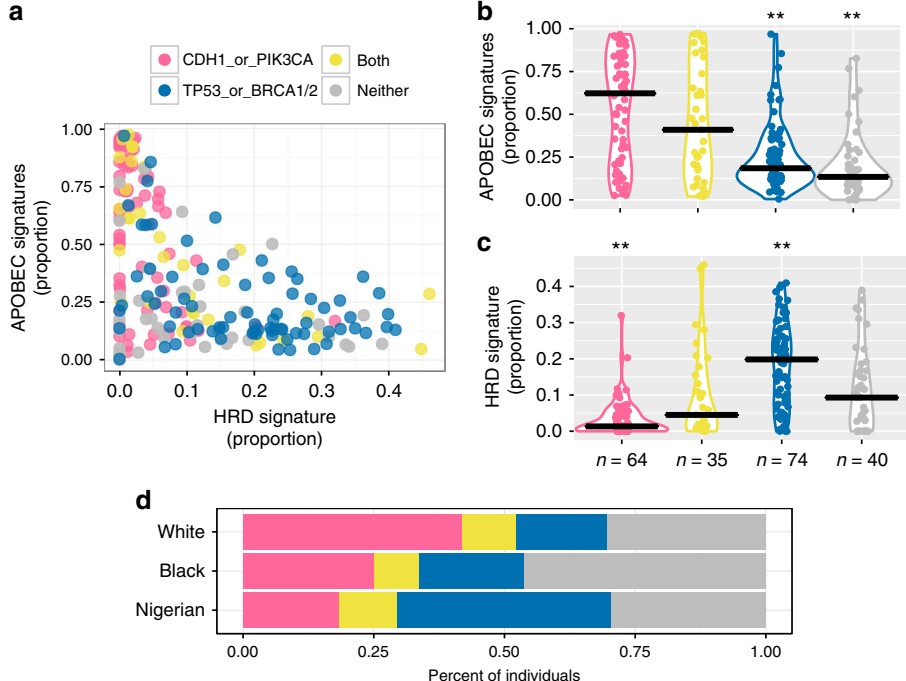

**Fig. 5** Driver gene mutations associate with APOBEC and HRD signature balance in HR+/HER2- breast cancer. **a** For each tumor, the proportion of APOBEC signatures (sum of APOBEC C > T and C > G) by the proportion of HRD is shown. Each patient is colored based on harboring a *CDH1* or *PIK3CA* mutation (pink), a *TP53* or *BRCA1/2* (including germline) mutation (blue), mutations from both aforementioned categories (yellow), or mutations in neither of the aforementioned categories (gray). These values are decomposed into violin plots for **b** APOBEC and **c** HRD signatures, respectively. Horizontal black bars represent the median contribution proportion for each group. Between group comparisons were made using a Kruskal–Wallis test followed by Dunn's test. Panels **a–c** were not restricted by race/ethnicity. **d** The proportion of HR +/HER2 − individuals falling into each mutational group by race/ethnicity (*n* White = 465; *n* Black = 80; *n* Nigerian = 27). This also includes samples for which mutation signatures were not estimated. **Groups that were significantly different (*P* < 0.05) from all three other categories

recurrent event in lung cancer and potentially affects response to anti-PD-1/anti-PD-L1 therapies[41]. Further studies to characterize the role for these genes in HER2 + tumors specifically and breast cancer in general are warranted.

The mutational landscape and signature patterns differed across racial/ethnic populations. In particular, the relatively younger Nigerian patients had more *TP53* and *GATA3* mutations than Blacks in TCGA, whereas both African ancestry groups had higher prevalence of these mutations than Whites. The frequencies of prognostically favorable *PIK3CA* and *CDH1* mutations were lower in women of African ancestry than in Whites, which may reflect differences in breast cancer risk factors across populations. Even when restricting to ER +/HER2 − breast cancer, tumors from Nigerian women were characterized by canonically aggressive molecular features, such as higher contributions from the HRD mutational signature, *TP53* mutations, and increased structural variation. Along with more pervasive HR negativity and HER2 positivity, the aggressive features of HR + tumors provide biological insight to why breast cancers in the unscreened and relatively younger female populations of West Africa are often fatal[42]. This study lays the foundation for a more concerted effort to reduce global disparities in cancer outcomes by first closing the knowledge gaps. Given the genomic landscape, Nigerian women would benefit from increased access to genomically tailored clinical trials and more effective treatments such as HER2-targeted therapy and PARP inhibition for HER2 + and HRD-deficient tumors, respectively[28].

There are certain limitations to this study including the relatively small sample size of Nigerian tumors and the fact that both TCGA and this study used convenient samples ascertained in Hospitals and may not reflect population rates. Nonetheless, this study underscores the need to include diverse populations when identifying and pursuing novel therapeutic targets[18]. It is possible that genetic and environmental factors not only drive subtype differentiation but also dictate evolutionary dynamics of a tumor. This latter assertion could help explain the observed mutational differences between racial/ethnic groups, a pattern which has also been noted comparing Black and Whites with colorectal cancer in the United States[43]. Similarly, strong associations between driver mutations and mutation signature contributions (e.g., *PIK3CA* and APOBEC signatures) pose a causality dilemma suited for further biological and epidemiological investigations. Overall, our results justify the need for future studies integrating germline and somatic genetics, as well as environmental factors, in order to better understand the root causes of disparities in breast cancer outcomes and develop more effective interventions to achieve health equity.

## Methods

**Biospecimen collection and pathological assessment**. This study was embedded within the Nigerian Breast Cancer Study (NBCS) and approved by the Institutional Review Board of all participating institutions. Patient ascertainment and details of the study have been previously published[2,44,45]. In collaboration with Novartis, NBCS was extended to Lagos State University Teaching Hospital (LASUTH). A grand total of 493 subjects were recruited from University College Hospital, Ibadan (UCH; *n* = 284) and LASUTH (*n* = 209) between February 2013 and September 2015. Each patient gave written informed consent before participation in the study. Six biopsy cores and peripheral blood were collected from each patient. Two biopsy cores were used for routine formalin fixation for clinical diagnosis and the remaining four cores were preserved in PAXgene Tissue containers (Qiagen, CA) for subsequent genomic material extraction. In addition, 27 mastectomy tissues were preserved in RNAlater. Complete pathology assessment was done central by study pathologists. Tumor burden was assessed based on cellularity, histology type, and morphological quality of tissue using TCGA best practices, and only tissues

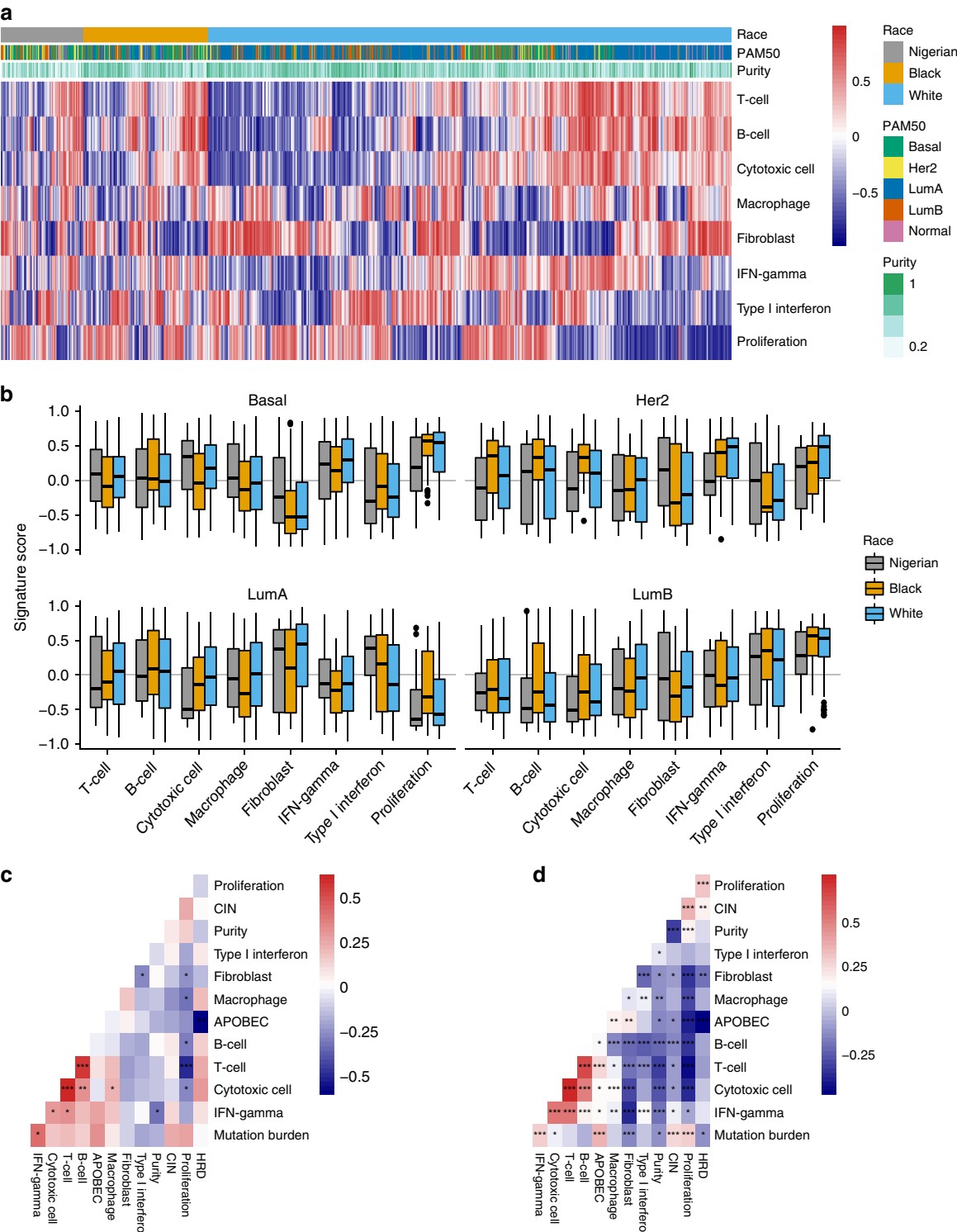

**Fig. 6** Gene signatures of immune cell infiltration. **a** Heatmap visualizing gene signature activation in all 1040 patients with RNA-seq data (Nigerian *n* = 103, Black *n* = 183, and White *n* = 754). High signature scores (red) indicate high overall expression of genes in the signatures, whereas low values (blue) indicate low expression. **b** Distribution of signature scores across PAM50 subtypes and ethnicities. **c**, **d** Pairwise Pearson's correlation of immune signatures as well as potential predictors of response to immunotherapy (APOBEC, HRD, CIN, mutation burden). The Nigerian data are shown in **c** and the combined Black and White cohorts in **d**. CIN chromosomal instability; HRD homologous recombination deficiency; IFN interferon. Each box represents the upper and lower quartiles of the data and the median is depicted with a horizontal line. Upper and lower whiskers extend to largest and smallest values within [1.5 × interquartile range], respectively. *$P$ < 0.05; **$P$ < 0.001, ***$P$ < 0.0001 (all adjusted using the Benjamini–Hochberg method)

containing 60% or more tumor cellularity were used for WGS. For WES, tissues containing 30% or more tumor cellularity were used. IHC on ER, PR, and HER2 were performed centrally in Nigeria and further reviewed in the United States. Cases with discordant results were again reviewed and resolved by the study pathologists. IHC scoring variables for Allred scoring algorithm were captured according to the 2013 ASCO/CAP standard reporting guidelines. Briefly, for ER and PR testing, immunoreactive tumor cells < 1% was recorded as negative and those with ≥ 1% were reported positive. All the positive ER and PR cases were graded in percentages stained cells and further scored in line with the Allred scoring system. Percentage of tumor staining for HER2 test were also reported along with a score of 0 and 1 + as negative, 2 + as equivocal, and 3 + as positive case. In addition, genomic copy number calls of HER2 and chromosome 17 ploidy were used as alternative to HER2 fluorescent in situ hybridization test. Overall, IHC calls were corroborated *ESR1*, *PGR*, and *ERBB2* expression using RNA-seq (Supplementary Fig. 1,1a–c).

**Sample selection and genomic material extraction**. Breast tumors were selected for sequencing following the TCGA guidelines[6]. Tumor samples containing > 60% tumor cellularity were selected for DNA extraction using PAXgene Tissue DNA kit (Qiagen). Gentra Puregene Blood Kit (Qiagen) was used to extract genomic DNA from blood. Extracted DNA were quality controlled for its purity, quantity, and integrity. Identity of the extracted DNA were tested using AmpFlSTR Identifiler PCR Amplification Kit (Thermo Fisher Scientific). Samples that match > 80% of the short tandem repeat profiles between tumor and germline DNA were considered authentic. RNA was extracted from PAXgene fixed tissues using the PAXgene Tissue RNA kit (Qiagen). RNA integrity (RIN) was determined for all samples by the RIN score given by the TapeStation (Agilent) read out. RNA samples that had RIN scores of 4 and above were included in downstream sequencing analysis.

**Next-generation sequencing data generation**. WES and RNA-seq were carried out at the Novartis Next Generation Diagnostics facility. Exome enrichment was performed on libraries (prepared by Illumina TruSeq Nano DNA Library Prep Kit) passing QC using Agilent SureSelect XT Human All Exon V4 baits and SureSelect XT capture enrichment reagents. Passing captured libraries are combined in equimolar pools with other captured libraries of compatible adapter barcodes. These pools were normalized with concentration and were sequenced on the Illumina HiSeq 2500 sequencer. Tumor samples had an average coverage depth of 139 × (63 × to 265 ×), normals 52 × (19 ×–205 ×; Supplementary Data 2a). WGS was performed at the University of Chicago High-throughput Genome Analysis Core (HGAC) and at the New York Genome Center (NYGC). Libraries were prepared using the Illumina Truseq DNA PCR-free Library Preparation Kit. Libraries were sequenced on an Illumina HiSeq 2000 sequencer at HGAC using 2 × 100bp paired-end format and HiSeq × sequencer (v2.5 chemistry) at NYGC using 2 × 150 bp cycles. Mean coverage depth tumor was at 98.5 × and normal was at 34.2 × (Supplementary Data 2b). For RNA-seq, total RNA were constructed into poly-A selected Illumina-compatible cDNA libraries using the Illumina TruSeq RNA Sample Prep kit. Passing cDNA libraries were combined in equimolar pools with other libraries of compatible adapter barcodes and later sequenced on the Illumina HiSeq 2500 sequencer. Average number of mapped reads per sample was 97 million (ranging from 36 to 232 million).

**Alignment of DNA sequence to reference genome**. For both exomes and genomes, reads were aligned to GRCh37 from GATK data bundle version 2.8 ([https://software.broadinstitute.org/gatk/]) using BWA-MEM (v0.7.12; [http://bio-bwa.sourceforge.net/])[46]. Duplicate reads were removed using PicardTools MarkDuplicates (v1.119; [https://broadinstitute.github.io/picard/]). Using a custom Fluidigm SNP panel, we confirmed that whole-exome BAM files matched the library DNA, to identify sample swaps in the sequencing lab or bioinformatics pipelines.

**Calling somatic single nucleotide variants**. Single-nucleotide variants (SNVs) were called using both MuTect (v1.1.7; [http://archive.broadinstitute.org/cancer/cga/mutect])[47] and Strelka (v1.0.13; [https://sites.google.com/site/strelkasomaticvariantcaller/])[48] with default parameters, except Strelka's depth filter was not used for exomes (isSkipDepthFilters = 1). Variants were called on the entirety of the genome in order to detect and retain any high-quality off-target calls. Any variant call that did not meet 'PASS' criteria for either algorithm was discarded. For a given tumor-normal pair, only SNVs called by both MuTect and Strelka were retained. Furthermore, using 1088 blood germline exomes (959 TCGA BRCA; 129 Nigerian), we constructed a panel of normal samples. For a given normal sample, a site needed to be covered by a minimum of ten reads to be included. Any SNV that was supported by 5% or more of reads (MAPQ (MAPping Quality) ≥ 20; Base quality ≥ 20) in two or more samples was removed. SNVs were later annotated with Oncotator ([http://archive.broadinstitute.org/cancer/cga/oncotator])[49] and those that met the required criteria ("COSMIC_n_overlapping_mutation > 1" AND "1000gp3_AF ≤ 0.005" AND "ExAC_AF ≤ 0.005") were considered likely to be somatic and were retained. This panel of normal process was also repeated for genomes (normal sample $n = 124$). All subsequent

SNV calls were annotated by Variant Effect Predictor (VEP) (v79; [http://useast.ensembl.org/info/docs/tools/vep/index.html])[50].

**Calling somatic insertions and deletions (indels)**. Small indels were called using Scalpel (v0.5.3; [http://scalpel.sourceforge.net/]) in somatic mode[51,52]. Variants were only called in known genic regions as defined by Broad.human.exome.b37.interval.bed from the GATK data bundle version 2.8. To minimize the number of false-positive calls, we employed the two-pass option. Default Scalpel filters were implemented, which required a minimum alternative allele count of four in the tumor, no alternative allele present in the normal, and a minimum tumor variant allele frequency of 5%. In addition, indel calls located in repetitive genomic regions (via DustMasker; [https://www.ncbi.nlm.nih.gov/IEB/ToolBox/CPP_DOC/lxr/source/src/app/dustmasker/]) or found in the 1000 Genomes Project Phase 3 release ([http://www.internationalgenome.org/]) were removed[53,54]. Finally, we implemented a pseudo panel of normals by aggregating all putative indel calls that failed Scalpel filters due to 'HighVafNormal' or 'HighAltCountNormal'. Any indel that failed in two or more samples was filtered. The remaining calls were annotated using VEP.

**Calling germline SNVs and indels**. For both exomes and genomes, reads were aligned to GRCh37 from GATK data bundle version 2.8. Duplicate reads were removed using PicardTools MarkDuplicates (v1.119). Both SNVs and indels were called using Platypus (v0.7.9.1; [http://www.well.ox.ac.uk/platypus]) in single-sample mode[55]. Only variants passing the Platypus 'PASS' filter were considered for downstream analysis. The resulting set of variants were annotated using VEP. All variants with an ExAC[56] allele frequency ≥ 0.05 were discarded. Remaining variants were considered deleterious if they were annotated as HIGH impact by VEP or a missense variant with a CADD[57] score > 25.

**Calling copy number alterations**. Allele-specific copy number in whole-exome data was called using PureCN (v1.7.16; [http://bioconductor.org/packages/PureCN/])[58]. Alternative purity and ploidy solutions were considered (Supplementary Fig. 3a–d). Genes were called amplified if the median exon copy number was ≥ 6 for focal gains ( < 3 Mb) or ≥ 7 for non-focal gains. Genes with median exon copy number of 0 were called lost. Non-focal amplifications of tumor suppressor genes were excluded[59]. As Affymetrix Genome-Wide Human SNP Array 6.0 data were available for the TCGA cohort, copy number calling was performed using ASCAT ([https://www.crick.ac.uk/peter-van-loo/software/ASCAT]; Supplementary Methods). Amplifications and deletions were called exactly as in the exome data. GISTIC (v2.0.22; [http://archive.broadinstitute.org/cancer/cga/gistic])[60] was used to identify significantly gained or lost genomic regions in the Nigerian cohort. TCGA GISTIC2 results were obtained from the BROAD FireBrowse portal ([http://firebrowse.org/]). CIN was defined as the fraction of the genome with copy number alteration. Details are provided in the Supplementary Methods.

**Calling SVs in WGS**. SVs (deletions, duplications, and inversions) were called with both Delly (v0.6.1; [https://github.com/dellytools/delly])[61] and Lumpy Express (v0.2.10; [https://github.com/arq5x/lumpy-sv])[62]. A panel of normal samples was constructed by taking all Delly SVs calls made in at least one ($n = 124$) normal sample, regardless of "PASS" or "LowQual" in the FILTER field. Any SV found within the panel of normals was removed from the analysis. All Delly SVs passing the aforementioned filters were queried within the matched Lumpy calls. Delly SVs corroborated by a Lumpy call (same SV type and breakpoints within 500 bp [up or downstream]) were retained. These consensus SVs were filtered if a breakpoint (from either Delly or Lumpy) fell within a repetitive genomic region according to DustMasker. Lastly, inversions were required to have split read evidence (at least one read) from both Delly and Lumpy.

**Estimating genetic ancestry of study population**. We estimated the ancestry of breast cancer patients from TCGA using principal component analysis as practiced by TCGA Analysis Working Group[36]. According to the estimated proportion of ancestry, patients were grouped into genomic Black ( ≥ 50% African ancestry), genomic White ( ≥ 90% European ancestry), and genomic Asian ( ≥ 90% Asian ancestry). All Nigerian patients were assumed to be 100% African with little to no admixture with other populations[63].

**Significantly mutated genes**. To detect significantly mutated genes we used MutSigCV (v1.4; [http://software.broadinstitute.org/cancer/software/genepattern/modules/docs/MutSigCV/])[23,64]. SNV and indel variant call formats from 1164 individuals were annotated with Oncotator using the oncotator_v1_ds_Jan262014 database. MutSigCV was then invoked with default parameters on the Oncotator generated MAF file. To reduce common false positives, we allowed only a single non-silent indel within a given gene per sample. Finally, for any gene to be called significantly mutated, we required it to have more than two individuals harboring non-silent mutations across the entire dataset.

**Mutation signatures in WES and WGS**. The Bioconductor ([https://bioconductor.org/]) package SomaticSignatures ([https://bioconductor.org/packages/SomaticSignatures/])[65] was used to estimate somatic mutational

signatures. The ability to reliably call mutation signatures depends on sufficient numbers of mutations. To this point, we used all high-quality exome SNVs, regardless of whether they are coding or non-coding. Any sample containing at least 100 SNVs was included for downstream assessment. In addition, in order to stimulate more accurate signature estimates, 122 WGS tumor-normal pairs were also included in addition to 500 WES pairs (Supplementary Data 1c). To account for variable mutation counts across samples, we used SomaticSignatures to normalize the mutation matrix before performing non-negative matrix factorization. We elected to estimate 9 signatures (Supplementary Fig. 4), as that was (1) consistent with the number of signatures identified previously in breast cancer ([http://cancer.sanger.ac.uk/cosmic/signatures]) and (2) as 9 signatures explained ~99% of variance when using 122 genomes alone. Using matrix algebra on the resulting exposure and mutation matrices, we calculated the relative contribution of the nine signatures on each sample. Contributions represent the proportion of mutations assigned to given mutation signature within each tumor (Supplementary Methods). Exomes were used for all mutation signature analyses unless explicitly stated.

**RNA-seq analysis and immune signatures**. Gene expression measurements were uniformly calculated using Omicsoft ArraySuite® software ([http://www.omicsoft.com/array-studio])[66] for Nigerian and TCGA samples. The RNA-seq reads passing quality control were aligned to the Human B37 genome. Read counts for the UCSC gene models were calculated by the software. The gene counts were upper quartile normalized with the edgeR Bioconductor/R package ([https://bioconductor.org/packages/edgeR/])[67] and batch normalized using ComBat as implemented in the sva package ([https://bioconductor.org/packages/sva/])[68]. Transcripts per million expression values were calculated based on the normalized counts. PAM50 classification was carried out using the pbcmc package ([https://bioconductor.org/packages/pbcmc/])[69] using the robust parameter. Nigerian PAM50 classifications were consistent with IHC calls (Supplementary Fig. 1,2). To characterize the immune and stromal microenvironment of these tumors, we assessed the expression of several pre-specified sets of immune and stromal cell gene expression markers (Supplementary Table 7). Gene signature scores were calculated using the GSVA R/Bioconductor package ([https://www.bioconductor.org/packages/GSVA/])[70].

**Statistical methods**. All statistical calculations were completed in in R. Names of the performed tests are provided in the text and all P-values are two-sided. Non-parametric tests were used when the underlying data types often lacked normality (e.g., mutation signature contributions). All boxplots throughout the manuscript are Tukey's style.

**Code availability**. SwiftSeq is available at [https://github.com/PittGenomics/SwiftSeq]

**URLs**. COSMiC, http://cancer.sanger.ac.uk/cosmic; Gene Ontology Consortium, [http://www.geneontology.org/]; ICGC, [https://www.genome.gov/10001688/]; SwiftSeq, [https://github.com/PittGenomics/SwiftSeq]; TCGA, [https://cancergenome.nih.gov/]

## Data availability

Raw TCGA data used in this analysis were downloaded from TCGA Data Portal or Cancer Genomics Hub, and their UUIDs are listed in Supplementary Data 1f–h. Access to the harmonized variant calls that support the findings of this study are available on request from the corresponding author (O.I.O.). The raw sequencing data from Nigerian cases is available through dbGaP under Study Accession phs001687.v1.p1.

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

## Acknowledgements

We are greatly indebted to all the patients who agreed to participate in this study and graciously donated their biological materials. We thank M. Meyerson, N. Solovieff, S. Jaeger, J. Monahan, D. Porter, A. Huang, S. Cameron, R. Schlegel, and M. Fishman for advice and discussion. This study was supported by U01-CA161032 awarded to D.H., K.W., and O.I.O.; Susan G. Komen for the Cure (SAC110026) and Breast Cancer Research Foundation grants awarded to O.I.O.; as well as by funding from the Novartis Institutes for Biomedical Research granted to O.I.O. and by the Francis Crick Institute that receives its core funding from Cancer Research UK (FC001202), the UK Medical Research Council (FC001202), and the Wellcome Trust (FC001202). Computational resources were provided by the Computation Institute and the Biological Sciences Division of the University of Chicago and Argonne National Laboratory, under grant 1S10OD018495-01. P.V.L. is a Winton Group Leader in recognition of the Winton Charitable Foundation's support toward the establishment of The Francis Crick Institute. We want to thank the New York Genome Center for the quality of the sequencing and analytic services provided. We also want to thank Qiagen for their generous donation of PAXgene Tissue Containers and DNA Extraction Kits for this study. This work is dedicated to the memory of the late Mrs. Anne Olorunde (Project Manager from LASUTH) and Professor Abideen Olayiwola Oluwasola (Lead Pathologist at University of Ibadan).

## Author contributions

K.P.W., D.H., O.I.O., and J.B. designed the experiments. A.O. and B.O. procured patient specimens and prepared genomic materials at UCH in Ibadan. They performed all immuno-histochemical analysis and preliminary Allred scores. O.A. served as Project Manager and coordinated study operation at UCH. C.B. and A.F. supervised laboratory staff and provided leadership for sample procurement, and DNA and RNA isolation at UCH. C.N., B.F., A.A., and T.O. recruited the patients and collected specimens from patients in Breast Clinic at UCH. O. Oluwasola, M.A., and A. Adeoye performed pathological assessment of patient specimens at UCH. O. Ojengbede served as Site-PI and provided overall supervision of the study at UCH. A.S. procured patient samples and performed pathological assessment at LASUTH. E.O. prepared tissue samples and performed all immuno-histochemical analysis and preliminary Allred scores at LASUTH. V. A., M.O., F.O., N.I., and A.P. recruited patients and collected specimens from patients in Breast Clinic at LASUTH. J.O. served as Site-PI and provided overall supervision of the study at LASUTH in Lagos. S.M., R.L., M. Macomber, and R.S.J. procured genomic materials and performed sequencing at NIBR. M. Morrissey and W.W. supervised the sequencing and data preparation at NIBR. D.P., B.W., and E.L. managed the study operation at NIBR. G.K. performed pathological assessment at the University of Chicago. W.C. procured patient specimens and assisted the pathological assessment at the University of Chicago. J.Z. prepared samples for genomic analysis at the University of Chicago. T.F.Y. and E.S. managed study operation at the University of Chicago. M.R., D. F., J.J.P., and A.J.G. assessed the quality of sequence data. J.J.P., M.R., Y.Z., T.F.Y., A.V., B.H., E.L., S.W., D.F., J.W., and D.H. analyzed the data. C.O.O. provided clinical research, research ethics, and data management training for all Nigerian investigators. D. H. and L.S.C. supervised the statistical analysis. K.P.W., D.H., O.I.O., and J.B. supervised data analysis and helped interpret results. J.J.P., M.R., Y.Z., T.F.Y., A.V., E.L., S.W., K.P. W., D.H., O.I.O., and J.B. wrote the manuscript. All authors reviewed and approved the final manuscript.

## Additional information

**Competing interests:** The authors declare the following competing interests: M.R., A.V., E.L., S.M., R.L., M.M., R.S.J., B.H., KM, M.P.M., W.W., D.P., B.W., and J.B. are all employees of Novartis Institutes for BioMedical Research. K.P.W. serves as President at Tempus. O.I.O is an equity stock holder of CancerIQ and Tempus. All other authors declare no competing interest.

Jason J. Pitt[1,17], Markus Riester [2], Yonglan Zheng [3], Toshio F. Yoshimatsu [3], Ayodele Sanni[4], Olayiwola Oluwasola[5], Artur Veloso[2], Emma Labrot[2], Shengfeng Wang[3,6], Abayomi Odetunde[7], Adeyinka Ademola[8], Babajide Okedere[7], Scott Mahan[2], Rebecca Leary[2], Maura Macomber[2], Mustapha Ajani[5], Ryan S. Johnson[2], Dominic Fitzgerald[1], A. Jason Grundstad[1], Jigyasa H. Tuteja[1], Galina Khramtsova[3], Jing Zhang[3], Elisabeth Sveen[3], Bryce Hwang[2], Wendy Clayton[3], Chibuzor Nkwodimmah[8], Bisola Famooto[8], Esther Obasi[4], Victor Aderoju[9], Mobolaji Oludara[9], Folusho Omodele[9], Odunayo Akinyele[3], Adewunmi Adeoye[5], Temidayo Ogundiran[8], Chinedum Babalola[7,10], Kenzie MacIsaac[2], Abiodun Popoola[11], Michael P. Morrissey[2], Lin S. Chen[12], Jiebiao Wang[12], Christopher O. Olopade[3], Adeyinka G. Falusi[7], Wendy Winckler[2], Kerstin Haase [13], Peter Van Loo [13,14], John Obafunwa[4], Dimitris Papoutsakis[2], Oladosu Ojengbede[15], Barbara Weber[2], Nasiru Ibrahim[9], Kevin P. White[1,16], Dezheng Huo[3,12], Olufunmilayo I. Olopade [1,3] & Jordi Barretina [2,18]

[1]Institute for Genomics and Systems Biology, University of Chicago, Chicago, IL 60637, USA. [2]Novartis Institutes for BioMedical Research, Cambridge, MA 02139, USA. [3]Center for Clinical Cancer Genetics & Global Health, Department of Medicine, University of Chicago, Chicago, IL 60637, USA. [4]Department of Pathology and Forensic Medicine, Lagos State University Teaching Hospital, Ikeja, Lagos, Nigeria. [5]Department of Pathology, University of Ibadan, Ibadan, Oyo, Nigeria. [6]Department of Epidemiology and Biostatistics, School of Public Health, Peking University Health Science Center, Beijing 100191, China. [7]Institute for Advanced Medical Research and Training, College of Medicine, University of Ibadan, Ibadan, Oyo, Nigeria. [8]Department of Surgery, University of Ibadan, Ibadan, Oyo, Nigeria. [9]Department of Surgery, Lagos State University Teaching Hospital, Ikeja, Lagos, Nigeria. [10]Department of Pharmaceutical Chemistry, University of Ibadan, Ibadan, Oyo, Nigeria. [11]Oncology Unit, Department of Radiology, Lagos State University, Ikeja, Lagos, Nigeria. [12]Department of Public Health Sciences, University of Chicago, Chicago, IL 60637, USA. [13]The Francis Crick Institute, 1 Midland Road, London NW1 1AT, UK. [14]Department of Human Genetics, University of Leuven, Oude Markt 13, Leuven 3000, Belgium. [15]Centre for Population and Reproductive Health, College of Medicine, University of Ibadan, Ibadan, Oyo, Nigeria. [16]Tempus Labs Inc., Chicago, IL, USA. [17]Present address: Cancer Science Institute of Singapore, National University of Singapore, 14 Medical Drive, Singapore 117599, Singapore. [18]Present address: Girona Biomedical Research Institute (IDIBGI), Girona 17007, Spain.

