## [Peer Review File · Nature Communications]

Reviewers' comments:

Reviewer #1 (Remarks to the Author):

This manuscript describes the characterization of 194 breast tumors in Nigeria at the IHC, RNA and DNA level, and compares it to data from TCGA. Authors report a number of differences across populations and speculate on the potential relevance of such differences.

The main strength of this report is providing the first relatively large and comprehensive characterization of breast tumors in women of West African ancestry. This is of interest as women of African ancestry tend to develop tumors with aggressive features and early age at onset. Comparisons across populations in this report, however, are limited due to the highly selected set of tumors in the Nigerian study as well as TCGA. Even if the cases included in the studies were representative of the underlying populations of women in Nigeria and the US, the large demographic differences as well as lack of population-based screening in Nigeria would make the interpretation of any reported differences in proportions (rather than age-specific incidences) of tumor features difficult. Given this limitation, findings from this report should be interpreted with more caution (a mention of this important limitation in the discussion is not enough, it also needs to be taken into account in the interpretation of findings).

Overall, it is difficult to know what is the main take-home message of this paper after the many comparisons of correlated features. Is the main message that, after accounting for differences in age at diagnosis and proportions of different IHC/PAM50 subtypes, tumors in the Nigerian study have different genomic features than TCGA-Black samples? And that these differences are limited to HR+/HER2- and HER2+ subtypes? To what extent do findings reflect different "biology" vs population differences in demographics and screening? The authors jump to conclusions assuming that differences reflect different tumor "biology" across populations. This seems premature given the data presented.

1. The title, abstract and discussion put emphasis on different findings: e.g. the title highlights the higher "rates" of HRD as the main finding, the abstract mentions HRD among several other findings, including four "novel" mutated genes, the discussion only mentions two of the four mutated genes and gives different emphasis to findings than the title or abstract.
2. I found quite striking the high proportion (no "incidence" as indicated in Figure 1) of HER2-enriched tumors in the Nigerian samples, while there is only a small difference in TN and no difference in basal subtypes for Nigerian compared to TCGA-Black samples. To what extent does this difference explain other findings?
3. Suppl Figure 3c and 3d show that Nigerian samples with the HER2+ subtype have lower tumor purity than TCGA samples of this subtype, or than Nigerian samples with other subtypes. Why is this? How can this have affected findings?
4. The HRD enrichment in Nigerian samples compared to TCGA samples is limited to HR+/HER2- and HER2+ subtypes. The higher percentage of this and other "aggressive" features in HR+/HER2- tumors in Nigerian cases seems an important finding, but this message is a bit lost amid several others. Are these patterns also seen when looking at PAM50 rather than IHC subtypes?
5. The authors talk about rates or incidences when they refer to proportions or prevalence, this should be fixed throughout the manuscript.
6. As shown in Figure 1, tumors have different combinations of data from WGS, WES and RNA-seq. Given that this implies that different number of tumors are included in different analyses, the actual number of tumors included should be added to ALL tables and figures throughout the manuscript. It should not just be the total number of tumors, but the numbers included in relevant comparisons (e.g. what are the number of tumors included in each of the boxplots in Figure 4? N's could be added to the bottom of each boxplot)
7. How do the IHC and mRNA data for the same genes compared in the Nigerian samples? Given the known relationships, it would be good to see how these compare as a measure of QC. Also, what is the concordance between the PAM50 and the IHC subtypes ?

8. How robust are the mutational signatures? Have the authors compared different available algorithms to evaluate robustness?
9. The legend for figures 4a and 4b seem switched. Would be good to label the y-axes indicating which one is HRD or signature 8.
10. The second page of the results mentions germline BRCA1/2 mutations, but this is not mentioned or shown elsewhere. It is also confusing that the first paragraph of the discussion talks about "heritable traits" and "genetically influenced" since the paper does not really address germline variation.

Reviewer #2 (Remarks to the Author):

Pitt et al applied genome, exome, and RNA sequencing, to examine the molecular features of breast cancers across 194 patients from Nigeria compared to 1,037 patients from The Cancer Genome Atlas (171 Black, 753 White, 113 other).

Here comes the first question/inconsistency in the results: it would have helped if the results were compared between the 194 patients and the groups of 171 Black, 753 White, 113 other in the TCGA throughout the entire analysis. The authors do so in some instances (when characterising the APOBEC signature) but not through the entire manuscript, which would have been, in my mind, most fair and informative as it will both create groups of equal size and will keep the ethnic differences separate.

For instance, when in the abstract it is reported that Triple Negative and HER2+ subtypes were enriched in Nigerians and had a greater prevalence of the homologous recombination deficiency (HRD) signature, higher TP53 mutation rate and increased structural variation, it is not clear compared to which group: the entire TCGA or the 171 Black in TCGA.

In the Chapter "Mutation landscape of Nigerians compared to Americans" data seem not to be compared, but the described separately for each of the groups. " Across all 1,164 individuals with WES data, we identified 25 genes that were significantly mutated above background"- this must refer to the entire TCGA dataset. More examples of ambiguity when it comes to which groups are compared to which appear further down in the text and all becomes only more complicated when additional stratification (lum/basal etc) is introduced within each ethnic group. If the text follows more faithfully the figures and the figure legends- it will be much more easy to follow.

To the reviewer the most natural presentation of the data will be for the Nigerian population first (if the paper is about those), frequency of mutations and within subclasses, then the relevant comparison to each ethnic group in the TCGA. If the authors claim that they have identified novel mutations from the TCGA dataset only, this should be in a separate chapter with emphasis on the mutation detection or other improvements that brought to this result and improvement compared to previous attempts.

The same criticism applies to the distinction between analysis of RNA and DNA seq. The two types of analysis are not clearly distinguished in the text of the paper and the reader has to follow carefully whether the authors talk about mutation or expression signatures, i.e. derived from DNA level or RNA level analysis. Not all possibilities to integrate these datasets are fully explored and not all transcriptional signatures - highlighted. Of course the motivation to look for immune signatures associated to mutational or copy number signatures is pertinent, but I am not sure if the conclusions reach that far as to identify some specific to the Nigerian population. However, one could look for other signatures for that.

These comments are addressed mainly to the main text and the presentation of the results there. They can be easily addressed with simple rearrangements and clarity of thought throughout the text with emphasis on the main message of the paper: the specificities of the Nigerian population and the commonalities that can be extended to other individuals of African descent. All figures, figure legends and supplementary material are evidence of rigorous work and point to the validity and the importance of this study, which in my view are great!

Response to reviewers

Pitt & Riester *et al.* - NCOMMS-17-26123

Reviewer 1

This manuscript describes the characterization of 194 breast tumors in Nigeria at the IHC, RNA and DNA level, and compares it to data from TCGA. Authors report a number of differences across populations and speculate on the potential relevance of such differences.

The main strength of this report is providing the first relatively large and comprehensive characterization of breast tumors in women of West African ancestry. This is of interest as women of African ancestry tend to develop tumors with aggressive features and early age at onset. Comparisons across populations in this report, however, are limited due to the highly selected set of tumors in the Nigerian study as well as TCGA. Even if the cases included in the studies were representative of the underlying populations of women in Nigeria and the US, the large demographic differences as well as lack of population-based screening in Nigeria would make the interpretation of any reported differences in proportions (rather than age-specific incidences) of tumor features difficult. Given this limitation, findings from this report should be interpreted with more caution (a mention of this important limitation in the discussion is not enough, it also needs to be taken into account in the interpretation of findings).

We appreciate the thoughtful suggestion of this reviewer. We are in complete agreement with the sentiments of the reviewer, which was the concern we had when we first published our findings on population differences on breast cancer subtypes from Nigeria in 2009¹. Since then, numerous publications have confirmed our initial report based on immunohistochemistry. We agree with the reviewer that samples from Nigeria and TCGA may not necessarily be representative of the diversity in the African Diaspora. However, this dataset represents the first relatively large and comprehensive characterization of breast tumors in Nigerian women (the largest country in Africa), which makes our findings novel, and thus we believe it should be compared with the overwhelming data available on women of European Ancestry in the TCGA. Of note, samples in Nigeria were consecutively collected from the two clinics in Ibadan and Lagos. Their age distribution (mean age about 50) is similar to that reported in population-based Ibadan Cancer Registry (mean = 48 years)², suggesting they are likely to be representative of women who present to oncology clinics for cancer treatment. It should also be noted that samples from TCGA are also convenient samples from multiple institutions. We have adjusted for age and molecular subtypes in the comparison of mutation landscapes to account for demographic differences across populations. Lack of screening in Nigeria significantly affects the distribution of tumor stage and would be expected to have less impact on tumor biology. Lifestyle, environmental, and genetic risk factors may play a bigger role in determining the mutation landscape in Nigerian patients as these mutations are cumulative events during carcinogenesis process, but that is beyond the scope of this initial report.

Overall, it is difficult to know what is the main take-home message of this paper after the many comparisons of correlated features. Is the main message that, after accounting for differences in age at diagnosis and proportions of different IHC/PAM50 subtypes, tumors in the Nigerian study have different genomic features than TCGA-Black samples?

Yes, this is the first study to show that Nigerian patients have genomic features that are different from TCGA-Blacks, which is to be expected given differences in geography, lifestyle and the level of admixture in African Americans. Nigerian women, an indigenous African population, regardless of IHC subtype, present with aggressive molecular features such as increased HRD mutation signature, increased prevalence of TP53 mutations, and greater structural variation. These are novel findings that were not immediately recognized in the large TCGA dataset. When differences in these genomic alterations were examined across racial/ethnic groups, they were mainly seen in HR+/HER2- tumors. Our group previously reported that HR (ER) positivity is a heritable trait³, and this study shows striking genomic differences in HR positive group although the Nigeria dataset was relatively small compared to the large TCGA dataset.

Within HR+/HER2- tumors

- HRD signature contributions were significantly higher in Nigerians compared to both TCGA Black and TCGA White cohorts
- TP53 mutation prevalence in Nigerians was greater than both TCGA Black and TCGA White groups
- The TCGA Black cohort had significantly more structural variation than the TCGA White cohort. Nigerians, in turn, had even greater structural variation than both of these groups.

In all subset analyses, Nigerians had higher prevalence of aggressive molecular features compared to both TCGA Black and TCGA White groups. With respect to structural variants, more events were also observed in the TCGA Black cohort than the TCGA White. We have improved the clarity of our story, and, throughout the revision, we have taken great care to ensure this theme is well-reflected in the title, abstract, results, and discussion.

And that these differences are limited to HR+/HER2- and HER2+ subtypes? To what extent do findings reflect different “biology” vs population differences in demographics and screening? The authors jump to conclusions assuming that differences reflect different tumor “biology” across populations. This seems premature given the data presented.

Demographic factors across populations could be a reason for the differences in mutation landscape and mutation signatures. To account for demographic factors, we have adjusted for age at diagnosis and molecular subtypes in the analysis, and the differences remained. Lack of screening in Nigeria can substantially affect the distribution of tumor stage but has less impact on tumor biology. This is reflected in our comparison of our findings in Nigeria with findings in the SEER database. Our data from Nigeria have implications for all non-European ancestry populations because lifestyle, environmental and genetic risk factors probably determine the differences in mutation landscape across populations. We acknowledge that the study findings should be replicated in larger studies with representative samples for diverse populations but this should be a global effort that is beyond the scope of this report.

1. The title, abstract and discussion put emphasis on different findings: e.g. the title highlights the higher “rates” of HRD as the main finding, the abstract mentions HRD among several other

findings, including four “novel” mutated genes, the discussion only mentions two of the four mutated genes and gives different emphasis to findings than the title or abstract.

We agree with this observation. For further specificity, we have changed the title to “Characterization of Nigerian Breast Cancer Reveals Prevalent Homologous Recombination Deficiency and Aggressive Molecular Features” to better reflect the take-home message of our manuscript. Rate has also been replaced by prevalence so as to use the correct terms (Review 1, comment 5 [below]). Additionally, we have now emphasized all four of the novel significantly mutated breast cancer genes in the discussion. This includes *GPS2*, which was reported as significantly mutated by the TCGA Pan-Cancer Working Group while this manuscript was under review⁴.

2. I found quite striking the high proportion (no “incidence” as indicated in Figure 1) of HER2-enriched tumors in the Nigerian samples, while there is only a small difference in TN and no difference in basal subtypes for Nigerian compared to TCGA-Black samples. To what extent does this difference explain other findings?

TCGA Breast Cancer samples were convenient samples from biobanks that were located in hospitals with breast Specialized Program of Research Excellence (SPORE) and hospitals that treat minority patients. The University of Chicago contributed samples to the TCGA and we acknowledge that these TCGA samples do not reflect population prevalence of HER2 positive tumors. In the US, most HER2 positive patients are treated with neoadjuvant chemotherapy and biospeciman banks do not have HER2+ tumors large enough for banking, whereas access to neoadjuvant chemotherapy is restricted in Nigeria. Thus, the proportion in Nigeria likely represents the true population prevalence of unscreened and untreated HER2+ tumors which is enriched in younger women. As discussed in **Supplementary Methods**, our Nigerian cohort was comprised of consecutive samples from two medical centers in Nigeria. Since TCGA applied its own unique collection methods, SEER data was included to serve as a population-based benchmark for comparison. The enrichment of HER2+ individuals from Nigeria compared to both TCGA and SEER is indeed striking but the Nigerian population is a decade younger than both TCGA and SEER population. It is difficult to completely rule out the possibility of oversampling from this IHC subtype because HER2+ subtype is aggressive and enriched in younger women. However, hypothetically, even if we oversampled HER2+ individuals from Nigeria, the comparisons within HER2+ tumors are still valid as statistical hypothesis testing for racial/ethnic differences was only performed within subtypes throughout the manuscript except the fourth paragraph of the results. Nonetheless, we agree with Reviewer 1 that scrutinizing the results for possible biases – here and elsewhere – is an important exercise. In accordance with Reviewer 1’s suggestion, we have interpreted the HER2+ enrichment in Nigerians more cautiously throughout the text considering possible issues regarding population structure as the population structure in Nigeria is vastly different from the population structure in SEER. Also, due to these interpretation nuances, we also no longer highlight HER2+ enrichment in the abstract since readers are not provided with sufficient context.

3. Suppl Figure 3c and 3d show that Nigerian samples with the HER2+ subtype have lower tumor purity than TCGA samples of this subtype, or than Nigerian samples with other subtypes. Why is this? How can this have affected findings?

Within Nigerians, the tumor purities of triple negative and HER2+ tumors are not statistically different from one another (**Supplementary Fig. 3c**). Lower purity in these two subtypes – which is also reflected in the TCGA samples – is, at least in part, due to greater lymphocytic infiltration⁵. Early onset breast cancer has been shown to have increased immune cell admixture⁶. With respect to comparing TCGA and Nigerians, general differences in purity can be explained by TCGA's more restrictive sample selection criteria. Any tumor with a pathology-based purity <60% was deemed ineligible for sequencing. Nigerian samples with pathological purities as low as 30% were subjected to exome sequencing. The latter point has been included within the main text methods section.

Thus, it is reasonable to conclude that data from our Nigeria cohort reveals novel findings that were possibly masked by the ascertainment bias in TCGA samples. There is paucity of data from non-European ancestry groups to make definitive conclusions. We, however, do not believe the tumor purity discrepancies between the TCGA and Nigerian cohorts had undue influence on our results, especially since our major findings were contained to the HR+/HER2- subtype. After much consideration, we were able to surmise one scenario where differences in purity could affect our results and conclusions. It has been reported that APOBEC-mediated mutations are more often subclonal⁷. If HER2+ Nigerian samples have fewer cancer cells, we could have less power to detect subclonal mutations, leading to under-sampling of APOBEC mutations. Since we are exploring the percent contribution of each mutation signature, a hypothetical drop in APOBEC signature representation could errantly increase the proportion of HRD signature activity in a sample. However, if this were the case we'd expect to see APOBEC signature differences amongst races/ethnicities in HER2+ tumors as well; however, this is not observed as depicted in **Supplementary Fig. 7**. Furthermore, analyses indicated that most exomes from HER2+ Nigerians were sufficiently powered to detect subclonal mutations. Following the power calculation by Carter et al., we find that 127 out of 129 have sufficient power (>0.8) to detect sub-clonality down to 50% of cells and 95 even down to 20%.

4. The HRD enrichment in Nigerian samples compared to TCGA samples is limited to HR+/HER2- and HER2+ subtypes. The higher percentage of this and other “aggressive” features in HR+/HER2- tumors in Nigerian cases seems an important finding, but this message is a bit lost amid several others. Are these patterns also seen when looking at PAM50 rather than IHC subtypes?.

We agree that the higher percentage of HRD and other “aggressive” features in HR+/HER2- tumors in Nigerian cases is an important finding. Due to resource limitation, we were unable to recapitulate the HRD finding using PAM50 subtypes, which was potentially a consequence of too few Nigerian samples having PAM50 data. To be utilized in the PAM50 version of this analysis, each sample needed to have at least 100 SNVs to reliably call mutation signatures and RNA sequencing data. These requirements resulted in only three and four Nigerian Luminal

A and Luminal B tumors, respectively. This is in contrast to 13 Nigerian HR+/HER2- tumors used for the IHC comparisons. However, as indicated in Supplementary Figure 12, we do observe expected concordance between IHC and PAM50 classifications in the Nigerian dataset.

5. The authors talk about rates or incidences when they refer to proportions or prevalence, this should be fixed throughout the manuscript.

We thank the reviewer for this correction. In epidemiological literature, “incidence” and “rate” have well-defined meanings. Throughout the revised manuscript the use of rate or incidence has been substituted by prevalence, proportion, occurrence, or pervasive[ness] when appropriate.

6. As shown in Figure 1, tumors have different combinations of data from WGS, WES and RNA-seq. Given that this implies that different number of tumors are included in different analyses, the actual number of tumors included should be added to ALL tables and figures throughout the manuscript. It should not just be the total number of tumors, but the numbers included in relevant comparisons (e.g. what are the number of tumors included in each of the boxplots in Figure 4? N's could be added to the bottom of each boxplot)

This is a helpful observation. Now, for each figure, the number of samples used in each group has either been explicitly added to the plot (e.g. **Figs. 4 & 5**) or stated within the figure legend (e.g. **Supplementary Fig. 6**).

7. How do the IHC and mRNA data for the same genes compared in the Nigerian samples? Given the known relationships, it would be good to see how these compare as a measure of QC. Also, what is the concordance between the PAM50 and the IHC subtypes ?

We thank Reviewer 1 for this incredibly helpful suggestion. We had previously conducted both of these analyses, though we did not include them in this manuscript. After further consideration, we are in agreement with Reviewer 1; these analyses would be useful to include as a quality control. *ESR1* expression by ER status, *PGR* by PR status, and *ERBB2* expression by HER2 status are shown in **Supplementary Fig. 11**. The IHC and PAM50 comparisons are depicted in **Supplementary Fig. 12**. All results indicate that Nigerian IHC subtype calls are consistent with expected⁸ gene expression and PAM50 patterns.

8. How robust are the mutational signatures? Have the authors compared different available algorithms to evaluate robustness?

We shared the same concerns as Reviewer 1 throughout our analyses because the signatures were called based on non-negative matrix factorization and the factorization is not unique. As shown in both the main text and supplementary methods (**Supplementary Figs. 5 and 6**), we were cautious to ensure that we only performed downstream analyses on reliable signatures.

- The mutation signatures themselves are robust based on the statistics and algorithms implemented by SomaticSignatures. As a precaution, we repeatedly inferred these

signatures five times and found that the calling of APOBEC, aging, signature 8, and HRD was consistent with each iteration.

- For a related and ongoing project, another analyst inferred mutation signatures using the samples from this manuscript as well as over 500 additional breast tumors. The resulting signatures, especially the ones discussed throughout our manuscript, were very consistent.
- While we only used one algorithm, we identified the same signatures as other groups (e.g. Broad Institute, Sanger Institute, etc.), which used their own in-house developed algorithms.

We must admit that a less pervasive signature could have been missed, but the reproducibility concern is exactly why we focused on signatures that correlated between exomes and genomes (**Fig. 5b**) and closely resembled signatures previously identified in breast cancer (**Fig. 5a**).

9. The legend for figures 4a and 4b seem switched. Would be good to label the y-axes indicating which one is HRD or signature 8.

We very much appreciate Reviewer 1's astute observation. We have confirmed that the labels within the Figure 4 legend were inaccurate. These details are now correct within this revision. We have also added the HRD and Signature 8 y-axis labels to **Fig. 4a-b**.

10. The second page of the results mentions germline BRCA1/2 mutations, but this is not mentioned or shown elsewhere. It is also confusing that the first paragraph of the discussion talks about "heritable traits" and "genetically influenced" since the paper does not really address germline variation.

While we certainly empathize with Reviewer 1's concern, the germline *BRCA1/2* results were included due to our emphasis on the HRD signature. Multiple studies have shown that known and predicted harmful alleles in these genes are associated with greater prevalence of the HRD signature, particularly in breast cancer. Had we failed to consider these alterations in our analysis, one could argue that they are responsible for the increased HRD activity in HR+/HER2- Nigerians. So we believe it is highly advantageous to keep this information within the results.

We agree with Reviewer 1 that the mention of "heritable traits" and "genetically influenced" are slightly out of context and require clarification. Our intention was to highlight evidence suggesting germline genetics can influence gene expression phenotypes important to breast cancer. Since our populations of interest have inherent genetic differences, it is reasonable to hypothesize that genetic background influences the somatic mutational landscape observed within these groups. We have added three sentences to the first paragraph of the discussion, which we believe helps present our line of reasoning more clearly.

Reviewer 2

Pitt et al applied genome, exome, and RNA sequencing, to examine the molecular features of breast cancers across 194 patients from Nigeria compared to 1,037 patients from The Cancer Genome Atlas (171 Black, 753 White, 113 other).

1. Here comes the first question/inconsistency in the results: it would have helped if the results were compared between the 194 patients and the groups of 171 Black, 753 White, 113 other in the TCGA throughout the entire analysis. The authors do so in some instances (when characterising the APOBEC signature) but not through the entire manuscript, which would have been, in my mind, most fair and informative as it will both create groups of equal size and will keep the ethnic differences separate.

We agree with Reviewer 2 that groups of consistent size would have been the ideal analysis structure; however, given our dataset, we considered this approach and found it to be unrealistic for two reasons, both of which relate to sample dropout due to missing information:

1. As depicted in Supplementary Figure 1, there was limited overlap of exome, RNAseq and WGS data types within Nigerians. If we required all of these data types to be present, the number of samples available would be reduced substantially. Even if we only required an individual to have exome and RNAseq data, this would have reduced the number of Nigerian individuals from 194 to 55. This – of course – would leverage only a fraction of our data.
2. Based on methodological limitations highlighted in previous reports⁹, we took a conservative approach and elected to perform mutation signature calls only on tumors with 100 or more SNVs, which cut our sample size down from 1,164 to 500.

Keeping consistent sample numbers throughout each analysis would have reduced our Nigerian and TCGA sample sizes by ~75% and ~50%, respectively. Due to our interest in multiple biological questions, we reasoned that the available sample approach would most effectively utilize this dataset, while the alternative approach could lead to numerous underpowered analyses. We do agree with Reviewer 2 that the number of samples used for each analysis requires more clarity. Also, per the suggestion of Reviewer 1, we have added the number of samples used in each comparison to main text and supplementary figures. We are hopeful that this help alleviate confusion resulting from the varying sample sizes used throughout our analyses.

2. For instance, when in the abstract it is reported that Triple Negative and HER2+ subtypes were enriched in Nigerians and had a greater prevalence of the homologous recombination deficiency (HRD) signature, higher TP53 mutation rate and increased structural variation, it is not clear compared to which group: the entire TCGA or the 171 Black in TCGA.

We thank Review 2 for this observation. The abstract has been rearranged, and the above findings have been stated as follows:

“Relative to Black and White cohorts, Nigerian HR+/HER2- tumors were characterized by increased homologous recombination deficiency signature, more pervasive *TP53* mutations and greater structural variation — indicating more aggressive biology.”

3. In the Chapter "Mutation landscape of Nigerians compared to Americans" data seem not to be compared, but the described separately for each of the groups. " Across all 1,164 individuals with WES data, we identified 25 genes that were significantly mutated above background"- this must refer to the entire TCGA dataset. (Response below in #4) More examples of ambiguity when it comes to which groups are compared to which appear further down in the text and all becomes only more complicated when additional stratification (lum/basal etc) is introduced within each ethnic group. If the text follows more faithfully the figures and the figure legends- it will be much more easy to follow. To the reviewer the most natural presentation of the data will be for the Nigerian population first (if the paper is about those), frequency of mutations and within subclasses, then the relevant comparison to each ethnic group in the TCGA.

We thank the reviewer for suggestions on how we can improve the clarity of our manuscript. It has stimulated multiple discussions amongst the authors. As detailed in point #4 below — as well as highlighted in the text — we clarified how the significantly mutated gene analysis was performed. From the outset, we only intended to contrast Nigerians, Blacks, and Whites using significantly mutated genes, which reduces the overall search space and multiple testing burden. Unfortunately, the sample size for the Nigerians was too small to perform a robust MutSig analysis on that cohort alone. We determined that it was necessary to assess significantly mutated genes across the entire dataset (TCGA and Nigerians). The resulting genes — along with significantly mutated genes discovered from previous studies — were utilized for all gene-centric, short variant analyses throughout the manuscript. We have added a terminal sentence to the second paragraph of the results to better clarify this objective. Despite multiple attempts to rearrange the text, we found that the Nigerian results are most relevant and succinct in the context of the other racial/ethnic cohorts. Opening with Nigerian genomes necessitated superfluous text in subsequent paragraphs and made it unclear why only certain genes were contrasted. We hope that our main text clarifications based on the reviewer's advice alleviates the aforementioned concerns.

4. If the authors claim that they have identified novel mutations from the TCGA dataset only, this should be in a separate chapter with emphasis on the mutation detection or other improvements that brought to this result and improvement compared to previous attempts.

The 1,164 samples included all TCGA samples as well as 129 exomes from Nigeria. We thought that the addition of Nigerians may provide sufficient power to discover new significantly mutated genes. As mentioned in the text, *PLK2* and *KDM6A* mutations were both enriched within HER2+ tumors. Given that the Nigerian tumors were enriched for this subtype, it's likely that these samples contributed to this new discovery. In the text we've now provided clarity by explicitly stating all samples, including Nigerians, were included when testing for significantly mutated genes.

5. The same criticism applies to the distinction between analysis of RNA and DNA seq. The two types of analysis are not clearly distinguished in the text of the paper and the reader has to follow carefully whether the authors talk about mutation or expression signatures, i.e. derived from DNA level or RNA level analysis.

We thank Reviewer 2 for this comment. We have edited the last two paragraphs of the results to be more explicit when RNAseq, mutation signatures, or immune signatures were being used.

6. Not all possibilities to integrate these datasets are fully explored and not all transcriptional signatures - highlighted. Of course the motivation to look for immune signatures associated to mutational or copy number signatures is pertinent, but I am not sure if the conclusions reach that far as to identify some specific to the Nigerian population. However, one could look for other signatures for that.

We agree with Reviewer 2. As mentioned in the text, racial/ethnic differences in immune signatures amongst populations were modest, but it should be noted that the field also suffers greatly from paucity of data from non-European ancestry groups like ours. There are more ways these data types can be integrated to elucidate differences, some of which are in progress. Due to financial resource constraints, we could not perform DNA and RNA sequencing on all the Nigerian tumors at hand. Given the novelty and complexity of the current results, we elected to explore additional integrative analyses via complementary studies. We're hopeful they will reveal more insight into the racial/ethnic disparities of breast cancer.

These comments are addressed mainly to the main text and the presentation of the results there. They can be easily addressed with simple rearrangements and clarity of thought throughout the text with emphasis on the main message of the paper: the specificities of the Nigerian population and the commonalities that can be extended to other individuals of African descent. All figures, figure legends and supplementary material are evidence of rigorous work and point to the validity and the importance of this study, which in my view are great!

We are in total agreement and are grateful that Reviewer 2 appreciates the validity and importance of this study.

References

1. Huo, D. *et al.* Population differences in breast cancer: survey in indigenous African women reveals over-representation of triple-negative breast cancer. *J. Clin. Oncol.* **27**, 4515–4521 (2009).
2. Jedy-Agba, E. *et al.* Cancer incidence in Nigeria: a report from population-based cancer registries. *Cancer Epidemiol.* **36**, e271–8 (2012).
3. Huo, D. *et al.* Comparison of Breast Cancer Molecular Features and Survival by African and European Ancestry in The Cancer Genome Atlas. *JAMA Oncol* (2017).
doi:10.1001/jamaoncol.2017.0595
4. Bailey, M. H. *et al.* Comprehensive Characterization of Cancer Driver Genes and Mutations. *Cell* **173**, 371–385.e18 (2018).
5. Stanton, S. E. & Disis, M. L. Clinical significance of tumor-infiltrating lymphocytes in breast cancer. *J Immunother Cancer* **4**, 59 (2016).
6. Thompson, E. *et al.* The immune microenvironment of breast ductal carcinoma in situ. *Mod. Pathol.* **29**, 249–258 (2016).
7. McGranahan, N. *et al.* Clonal status of actionable driver events and the timing of mutational processes in cancer evolution. *Sci. Transl. Med.* **7**, 283ra54 (2015).
8. Nielsen, T. O. *et al.* A comparison of PAM50 intrinsic subtyping with immunohistochemistry and clinical prognostic factors in tamoxifen-treated estrogen receptor-positive breast cancer. *Clin. Cancer Res.* **16**, 5222–5232 (2010).
9. Alexandrov, L. B., Nik-Zainal, S., Wedge, D. C., Campbell, P. J. & Stratton, M. R. Deciphering signatures of mutational processes operative in human cancer. *Cell Rep.* **3**, 246–259 (2013).

REVIEWERS' COMMENTS:

Reviewer #1 (Remarks to the Author):

I would like to thank the authors for the thoughtful responses to my comments and the revisions that have increased the clarity of the manuscript.

Reviewer #2 (Remarks to the Author):

The abstract and the text of the paper (both results and discussion) have been re-written to meet my concerns. I agree and understood that the addition of Nigerian samples to the TCGA provides additional power to discover new significantly mutated genes, but still maintain that in the next step, comparison between the populations, the Nigerian set should be discussed distinctly and separately, if this should be the topic of the paper. I see also that the authors have edited the last two paragraphs of the results to be more explicit when RNAseq, mutation signatures, or immune signatures are used.

Response to reviewers

Pitt & Riester *et al.* - NCOMMS-17-26123 – second review

Reviewer 1

I would like to thank the authors for the thoughtful responses to my comments and the revisions that have increased the clarity of the manuscript.

We thank this reviewer as his/her comments as they significantly improved our manuscript.

Reviewer 2

The abstract and the text of the paper (both results and discussion) have been re-written to meet my concerns. I agree and understand that the addition of Nigerian samples to the TCGA provides additional power to discover new significantly mutated genes, but still maintain that in the next step, comparison between the populations, the Nigerian set should be discussed distinctly and separately, if this should be the topic of the paper. I see also that the authors have edited the last two paragraphs of the results to be more explicit when RNAseq, mutation signatures, or immune signatures are used.

We are glad to hear our revisions have addressed the concerns of this reviewer. During comparisons amongst populations, the Nigerian dataset is consistently presented distinctly and separately from TCGA. However, since previous clinical and epidemiological studies suggest that unknown factors contribute to racial/ethnic disparities in incidence and outcomes of breast cancer, we contrasted White and Black cohorts from TCGA when presenting the results from Nigerians. As acknowledged by this reviewer above, we also integrated all racial/ethnic cohorts for one initial analysis to improve detection power for significantly mutated genes, which were subsequently used to reduce our genic search space. Both first authors and three senior authors independently attempted to rearrange the manuscript so the mutational landscape of the Nigerian cohort is first described in its entirety and then compared to TCGA datasets second. Despite these efforts, we could not earnestly accomplish this restructuring without decreasing clarity and increasing redundancy of the manuscript. So while we are emphatic to this reviewer's preference, we believe that retaining the current manuscript structure is the most coherent way to disseminate our results to the wider community. Despite this disagreement, we are very thankful for this suggestion as it caused us to critically analyze, assess, and improve our manuscript.